# The Sustainability of International Accreditations and Their Impact on Students' Choices in Selecting the Universities

**Cristina Fleșeriu [1], Florin Sebastian Duma [2], Ioan Alin Nistor [3] and Dragoș Păun [3,*]**

[1]  Department of Hospitality Service, Faculty of Business, Universitatea Babeș-Bolyai, 400174 Cluj-Napoca, Romania; cristin.fleseriu@tbs.ubbcluj.ro

[2]  Department of European Studies and Governance, Faculty of European Studies, Universitatea Babeș-Bolyai, 400174 Cluj-Napoca, Romania; florin.duma@ubbcluj.ro

[3]  Department of Business, Faculty of Business, Universitatea Babeș-Bolyai, 400174 Cluj-Napoca, Romania; ioan.nistor@tbs.ubbcluj.ro

*   Correspondence: dragos.paun@tbs.ubbcluj.ro; Tel.: +40-264-599170

**Abstract:** The purpose of the current article is to determine the sustainability of international accreditations for business schools. As international accreditations are viewed as a costly process, universities must think if this endeavor could have a positive impact in the long run. From an impact point of view, we look at the intake of students, focusing on the factors that impact the decision of students in their choice of university. We have noticed that these international accreditations are pursued by business schools to increase their outreach and to receive a certification of quality that is recognized overseas. We consider the hypothesis that international accreditation is a key factor in the decision-making process of candidates, and we tested it by applying a questionnaire to 400 business and economics students that are studying in two business schools. From the 400 students that answered the questionnaire, only 199 responses were considered fully answered and proper for our study. Our results show that there is a difference between French and Romanian students in the choice of universities. While both groups agree that internationalization is important, their decisions are based on different elements. Our research is among the few that look both at the student choice and at the impact of the international accreditation on the student numbers.

**Keywords:** AACSB; quality; business school; decision-making process

## 1. Introduction

In the last years, there has been an ongoing discussion related to classification and quality in higher education with several entities emerging in order to either classify the higher education institutions or to accredit them. From a quality perspective, we have several viewpoints. Harvey and Green [1] presented different concepts of quality as perceived by the various stakeholders in higher education. According to them, stakeholders' insights on quality could be classified based on five definitions of quality: exceptional, perfection, fitness for purpose, value for money, and transformation. Other studies show that competence of academic staff was previously shown to be the most important dimension of quality by undergraduate students according to Munasinghe and Rathnasiri [2], but the study Dicker et al. [3] concludes that personal qualities rather than the specific competence of academic staff were highlighted by students. We also noticed that quality might be defined in different ways based on country or culture. Finding a universal definition of quality is not easy because of geography and cultural differences, but also because of quantification difficulties and, last but not least, due to the fact that it is not always an objective process. Moreover, quality is a dynamic concept that is

changing over time [4]. Therefore, when defining quality in higher education, we need to take into account the main groups of stakeholders: the providers (the community, the taxpayers, etc.), the users of products (the students), the beneficiaries of the output (the employers), and the employees of the sector (professors, administrators, etc.) [5–7]. Each group has a different interpretation of quality, yet all of them should be taken into consideration. Due to the fact that it is difficult, if not impossible, to find a universally accepted definition, we can broadly classify these as stakeholder-driven and standard-driven definitions on one hand, and indicator-related definitions reflecting the desired inputs and outputs, on the other hand [5].

Due to this variety of definitions, there is a demand to have a common measure of quality in higher education. This common measure of quality has been given in recent years by different organizations that either rank universities or give international accreditations. Among the most famous university rankings, we can name the Academic Ranking of World Universities, Times Higher Education Ranking, FT Business School Rankings, QS World Rankings, U-Multirank, and Center for World University Rankings. All of these rankings are based on different indicators and are published by private entities; thus, their methodologies are frequently under scrutiny.

International organizations have also sought to fill the gap in determining the quality of universities and started certifying the quality of universities by implementing different membership or accreditation processes. Most of these associations or entities are in fields that have high mobility of international students and where there is a need for a global system of quality certification that students would recognize. The demand for global quality assurance in the field of law would be limited due to the domestic nature of the studies unlike the case of economics where we see the biggest international movement. Universities must also face the challenges and opportunities of borderless higher education, e-learning, and the increase in mobility programs.

Focusing on business economics, we see that in recent years there are more and more associations that are linked with memberships and accreditations. International accreditations can be split into professional accreditations and academic accreditations. The first category of professional accreditations is mostly linked with professional bodies or companies that are using this tool as a recruiting mechanism and are offering students from accredited schools different exemptions for exams or fee reductions. Among professional accreditations, we have the ACCA Exemption Accreditation [8], CIMA [9], CFA University Affiliation Program [10], and CIM [11].

Academic accreditation has always been a national issue due to the fact that universities were required, in the past, to meet national needs and to respect domestic regulations. The increase in student mobility and the introduction of the European Credit Transfer System have created a need for the standardization of national accreditations and thus, we have seen that a series of voluntary associations have emerged. ENQA (European Association for Quality Assurance in Higher Education) is such a body that incorporates different national quality assurance organizations and organizes conferences, workshops, and seminars as well as international quality assurance projects and cooperation activities with stakeholders. The United States has a different system and one of the longest traditions in accreditations and accreditation bodies [12]. It is in the US that the biggest international accreditation organization was established, i.e., AACSB. AACSB started out in 1916 as the American Assembly of Collegiate Schools of Business and accredited only US institutions until 1968. Currently, AACSB stands together with EFMD (European Foundation for Management Development) and AMBA (Association of MBAs) on top of the international business and economics accreditations. Other international accreditation bodies are NIBS (Network of International Business Schools), CEEMAN (Central and East European Management Development Association), ECBE (European Council for Business Education), and FIBAA (Foundation for International Business Administration Accreditation).

The current paper focuses on the sustainability of international accreditation given that the accreditation process is considered a costly process [13,14]. The main question that our study wants to answer is linked to the financial sustainability of international accreditation and the profitability of such an endeavor. Linked to this, the current study wants to see if international accreditation is

a factor for students' choice in programs and if there are regional differences between France and Romania. The financing of universities is mostly linked to endowment, state subsidies, or state funding and tuition [15–18]. For most public universities an increase in revenue can come from an increase in student numbers. Romanian universities rely more on student numbers, as private money or sponsorships are almost irrelevant. Looking at the preference of students and based on students' traditional habits, we have formulated our first hypothesis, which is, "Students select universities based on national rankings and employability." Our second hypothesis, which derives from the first one, is that "The international dimension of universities does not play a role when candidates select their desired program." The first two hypotheses were based on tradition and status-quo so we decided to include a third hypothesis based on the idea of our study, i.e., "The international accreditation obtained by Transylvania Business School and the membership in AACSB have led to an increase in student numbers." The last hypothesis subsequently leads to financial performance as student numbers lead to an increase in revenues [19,20].

Our research is looking at sustainability from two perspectives: the expense part, due to the fact that most international accreditations are a costly process, and the revenue part, which is driven mainly by the intake of students. Thus, in order to also look at the potential intake of students, we decided to look also at the business students' perception of international accreditations.

The most important international organization, AACSB, had, according to the data available in October 2019, 1610 global members (47% in the Americas, 23% in the Asia Pacific, 29% in the EMEA) and 856 accredited institutions (67% in the Americas, 16% in the Asia Pacific, 17% in the EMEA). The low number of accredited schools in the EMEA region could be explained by the fact that, as mentioned before, most institutions have relied on their national accreditation and have recruited their students based on this national quality assurance indicator. However, changing demographics, a lower number of high school students in the European Union member states, and student mobility have created a need for international validation and quality recognition. As such, if we look at the current universities in the AACSB accreditation process, we notice that EMEA universities represent the majority. From the 267 schools in the accreditation process, 21% are based in the Americas, 30% in the Asia Pacific, and 49% in the EMEA.

International accreditations are a certification of quality but the cost of obtaining them has prevented most universities from applying for them. For most of these international accreditations, the costs involved are divided into two categories: fees for obtaining the accreditation and fees that are to be paid by the accredited institutions after receiving the accreditation. In some cases, the costs are lower (e.g., for NIBS accreditation), but in the case of the so-called "triple crown" accreditation (AACSB, EQUIS, and AMBA) they can be quite prohibitive.

## 2. Literature Review

We have found a limited number of studies that are related to students' choice of university and even fewer studies focusing on the economic impact of international accreditations [13,21–25]. Most of these studies either focus on domestic vs. foreign or are related just to the students' choice without mentioning AACSB or any other international accreditation. Tamtekin Aydın [26] published a literature review on the university choice process and among the main factors, family, reference groups, reputation, location, job prospects, personal factors, and financial factors such as cost or financial aid were identified. Nemar and Vrontis [27] conducted a similar analysis focusing on Lebanon but their goal was to research the students' choice in order to help universities to segment their customers and to better target their marketing efforts. Ahmad and Buchanan [28] conducted a study to see why students chose to study at international campuses in Malaysia. The findings of their survey, deployed on 218 students, show that institutional and academic reputations, the marketability of the degree, the low tuition fees compared to home institutions, the low cost of living, the label of a safe country for study, the similarity of education systems as well as cultural proximity account for factors in the decision-making process. Winter and Chapleo [29] focused their exploratory study on students

from the United Kingdom and revealed that the first impression and contact between university and students is fundamental in the latter's choice of studies. McManus et al. [30] also focused their study on the UK but theirs are important factors that help students decide in their search for a higher education institution. Walsh and Cullinan [31] conducted a similar analysis focusing on Ireland and looking at factors such as peers, siblings, and parental influences. Parental guidance and professors are the main reasons determining university choice also in China, based on the results of a study by Liu and Morgan [32]. Casidy and Wymer [33] clustered the students based on their choices and showed that one of the clusters is represented by "prestige-seeking innovators". The students that were included in this cluster would always have a positive view of their choice of university, while the students in the "strivers" cluster would regret their choice and seek to change it to a more prestigious university, revealing that the prestige of universities plays an important role in students' choice. Chang et al. [34] focused on the impact of the AACSB Accreditation on business school students in Taiwan. The results of their study showed that there is a positive effect on students' effectiveness of learning, organizational identification, organizational citizenship behavior, and learning satisfaction. The authors concluded that the accreditation is a guarantee for quality.

Ke et al. [35], Azad and Seyyed [36] focused their studies on the impact of the AACSB accreditation on research productivity. The study analyzed four accredited universities by looking at the research performance prior to and after receiving the accreditation. They concluded that AACSB accreditation has a positive impact on research productivity. Elliott [23] analyzed the impact of accreditation and concluded that AACSB accreditation has a high impact on enhancing reputation. Webster and Hammond [37] presented the results of a national survey examining the levels of reported customer and market orientation toward students and explored their impact on organizational performance. The results of the study are that customer and market orientation do indeed affect organizational performance. Teixeira and Maccari [38] carried out a systematic literature review of the institutional role of school accreditation and they found that there is a duality to these institutional accreditations. On the one hand, they provide a quality seal and on the other hand, they also provide legitimacy. Looking at the impact of the accreditation on the career of the graduates, studies reveal that there is limited evidence, with [39] mentioning that graduates of accredited schools do not outperform the ones from non-accredited schools, but findings [40] state that the status of an accredited school gives graduates an advantage.

As a result of the literature review, we concluded that a study that focuses on the financial impact and therefore, on the sustainability of international accreditations would be a valuable addition to the literature. To look at the international context, we decided to focus our study on the choice of students from Romania and France because the financial impact is clear—international accreditation generates the same costs irrespective of the country one is located in—but the difference between French and Romanian students' choice of university is needed and would bring novelty to the literature. The international context is also clearly making a high impact, as higher education institutions find themselves in a struggle for funding and, therefore, aim to expand their reach [24,41,42].

## 3. Data Source and Method

Our study focuses on the main business schools located in the EMEA region in Romania and looks at the sustainability of international accreditation. As such, the main question that our study wants to answer is linked to the financial sustainability of international accreditation and the profitability of such an endeavor. Linked to this, the current study wants to see if international accreditation is a factor for students' choice in programs and if there are regional differences between France and Romania.

We selected several methods to test our hypotheses. We deployed a questionnaire, available in the Annex of this paper, that looked at the students' choices for universities in Romania but as well as France. We also selected French students to see if there is a regional bias in the selection criteria of students. The participation of students in the questionnaire (Appendix A) was voluntary, and the questions were sent to random students from all types of programs and study years via electronic platforms.

The questionnaire was sent to a total of 729 students from both business schools. The responses were voluntary and anonymous, and we received 400 responses (54.86% response rate) but only 199 questionnaires were validated and considered for this study (27.29% of total questionnaires sent). Transylvania Business School (Faculty of Business) within Universitatea Babes-Bolyai of Cluj-Napoca, Romania was selected because it is the only internationally accredited business/economics school in Romania (2015 NIBS accreditation) and the only one in the process of obtaining the AACSB accreditation, having started in 2018. Transylvania Business School is part of the biggest and most comprehensive public university in Romania and is financed mostly from the public budget. The second university that we selected is Ecole de Management de Normandie (EM Normandie), a French business school with campuses in Normandie (Caen and Le Havre), Paris, Oxford, and Dublin. The French business school was selected as the majority of accredited schools in the EMEA region come from France. When looking at the size of the two business schools we notice that they have a similar position within their country. By size, they would traditionally be classified as mid-sized business schools with a strong regional impact. EM Normandie is accredited by AACSB and also holds EQUIS and EPAS accreditations from EFMD.

As mentioned before, international accreditations are a costly process but they are, at the same time, different. For some institutions, accreditation involves a benchmarking process (EFMD, NIBS, CEEMAN) while others, like AACSB, involve a process of converging to the standards of the organization. The NIBS accreditation that Transylvania Business School received in 2015 involved a three-stage process that included a self-assessment questionnaire focusing on the international dimensions of the business school, a visit from a peer-review team, and a decision from the NIBS Board. Once the board and the peer-review team assessed that the benchmarks were attained, Transylvania Business School received the accreditation. The AACSB accreditation process is not a benchmark-type process due to the nature, complexity, outreach, and scope of the organization. The accreditation is a journey that comprises an eligibility application, an initial self-evaluation report, and a peer-review visit. In order to receive the AACSB accreditation, each business school must prove that it meets the requirements set by the Accreditation Committee.

## 4. Results

To test the first and second hypotheses and the difference between the choice of the Romanian and foreign students, we applied a questionnaire to students studying both at the bachelor and master levels. The sample comprised 60 foreign students and 139 Romanian students. The gender ratio was quite balanced, with 53% female respondents and 47% male respondents for the foreign students, and 62% female respondents and 38% male respondents for the Romanian students (Figure 1).

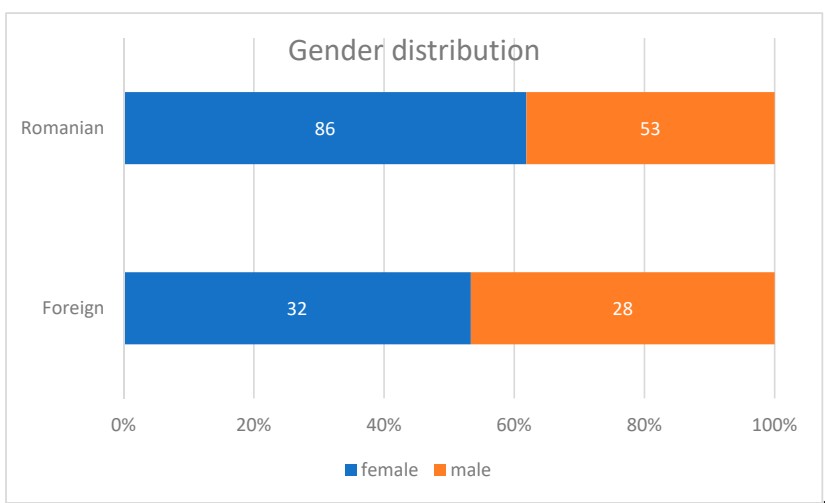

**Figure 1.** Gender distribution of the students.

The respondents came from both study programs (bachelor's and master's) from most study years. The gender balance is representative and follows the pattern of the schools, both of them reporting more female students than male. Their distribution can be seen in Table 1.

**Table 1.** Distribution of students based on the level of education.

| Study Program/Year | Foreign | | Romanian | | Total |
|---|---|---|---|---|---|
| | Frequency | % | Frequency | % | |
| Bachelor | 36 | 21.56% | 131 | 78.44% | 167 |
| 1st year | 0 | 0.00% | 61 | 100.00% | 61 |
| 2nd year | 6 | 10.71% | 50 | 89.29% | 56 |
| 3rd year | 26 | 56.52% | 20 | 43.48% | 46 |
| 4th year | 4 | 100.00% | 0 | 0.00% | 4 |
| Master | 24 | 75.00% | 8 | 25.00% | 32 |
| 1st year | 10 | 62.50% | 6 | 37.50% | 16 |
| 2nd year | 14 | 87.50% | 2 | 12.50% | 16 |
| Grand Total | 60 | 30.15% | 139 | 69.85% | 199 |

In order to test hypothesis three, the impact of the international accreditations on the recruitment of Transylvania Business School, we analyzed the student enrollment in both bachelor's and master's programs in the last eight years. When looking at the enrollment, we looked at the final number of students registered in the university programs (based on data from the beginning of the academic year) for both full-time and exchange students. Based on this, we can also evaluate the financial impact of international accreditation by looking at the turnover generated from student tuition. Public universities in Romania have two sources of funding when it comes to students (government-based or tuition-free and tuition fees). We did not look at the total turnover of the business school because, in these eight years, we experienced a change of students' per capita funding from the budget. Fortunately for our research, Transylvania Business School has had almost a constant number of students whose seats have been funded by the state, thus we can consider only the impact of the changes in the number of tuition students. Table 2 splits the students into two categories—students that pay a tuition fee which is valid for Romanian and EEA students, and students from outside the EEA region.

**Table 2.** Universities where students are enrolled.

| University | Foreign | Romanian | Total |
|---|---|---|---|
| Transylvania Business School | 7 | 139 | 146 |
| EM Normandie | 44 | 0 | 44 |
| Other | 9 | 0 | 9 |
| Total | 60 | 139 | 199 |

As the questionnaire focuses on students from EM Normandie and Transylvania Business School, the majority are from these institutions. Students who answered "other" are mainly exchange students who, in the academic year 2019–2020, were in their abroad semester at one of the two business schools.

Table 3 presents the most cited determinants in the choice of university, which, in the case of Romanian students, is the program in which they are enrolled, while for foreign students it is the international accreditation of the university. The results clearly contradict our hypothesis. Our first hypothesis was that students select their universities based on employability and national rankings, but as the results show these options were among the last. Romanian students look at the program where they study and do not look necessarily at the international dimension. We also see a clear difference between Romanian and French students and we believe that international accreditation is important for the latter group because most French Business schools offer a mandatory exchange semester so students are aware of this when joining the school, increasing the importance of internationalization.

**Table 3.** Determinants for choosing the university.

| Main Reason for Choosing the University | Foreign | | Romanian | | Total |
|---|---|---|---|---|---|
| | Frequency | % | Frequency | % | |
| The program where I am enrolled | 21 | 35.00% | 69 | 49.64% | 90 |
| International accreditation of the business school/university | 25 | 41.67% | 38 | 27.34% | 63 |
| City | 3 | 5.00% | 15 | 10.79% | 18 |
| Friend's recommendation | 3 | 5.00% | 8 | 5.76% | 11 |
| Recommendation of the family | 2 | 3.33% | 6 | 4.32% | 8 |
| Tradition of the business school/university | 2 | 3.33% | 3 | 2.16% | 5 |
| Employability chances | 4 | 6.67% | - | - | 4 |
| **Total** | **60** | **100%** | **139** | **100%** | **199** |

Table 4 presents the most important element when choosing a program. From the point of view of both foreign and Romanian students, it is the employability after graduation, while the city is less important. We can notice that international accreditation is ranked 3rd by the students, even before the internationalization and the national accreditation. These results are in line with our second hypothesis which mentioned that the international dimension is not important for student's choice. However, noticing the cultural context, while 20.86% of the Romanian students consider internationalization (accreditation and relations) important, the percentage increases to 40% for French students. As mentioned before, the mandatory study abroad could be a reason for this high percentage but one must also see the financial challenges that studying abroad brings. Surprisingly, the city is ranked last even though Cluj-Napoca, former European Youth Capital, is viewed as one of the most developed and attractive cities in Eastern Europe by young people.

**Table 4.** The most important elements when choosing a program.

| The Most Important Elements When Choosing a Program | Foreign | | Romanian | | Total |
|---|---|---|---|---|---|
| | Frequency | % | Frequency | % | |
| Employability after graduation | 17 | 28.33% | 59 | 42.45% | 76 |
| Content of the program | 15 | 25.00% | 50 | 35.97% | 65 |
| International accreditation | 13 | 21.67% | 17 | 12.23% | 30 |
| International relations | 11 | 18.33% | 12 | 8.63% | 23 |
| National accreditation | 2 | 3.33% | 1 | 0.72% | 3 |
| City | 2 | 3.33% | 0 | 0.00% | 2 |
| **Total** | **60** | **100%** | **139** | **100%** | **199** |

Table 5 presents the main reasons for choosing a university and the most important aspects when choosing a university, which were compressed into the following categories: university, location, employability, and recommendations (in the case of the reason for choosing a university).

A chi-square test of independence was performed to examine the relationship between gender and the main reason for choosing a university. The relationship between these variables was not significant, $X^2$ (3, $N$ = 199) = 0.368, $p$ = 0.947. There was no significant difference between men and women as to the reason for choosing a university. The same was true for the importance of choosing a university, $X^2$ (2, $N$ = 199) = 4.253, $p$ = 0.119.

A second chi-square test of independence was performed to examine the relationship between study programs (bachelor's or master's) and the main reason for choosing a university. The relation between these variables was not significant, $X^2$ (3, $N$ = 199) = 4.181, $p$ = 0.243. There was no significant difference between bachelor's and master's students in the reason for choosing a university. The same was true for the importance of choosing a university, $X^2$ (2, $N$ = 199) = 0.674, $p$ = 0.714.

A third chi-square test of independence was performed to examine the relationship between student origin (foreign or Romanian) and the main reason for choosing a university. Although the

relationship between these variables was significant, $X^2$ (3, $N$ = 199) = 10.957, $p$ = 0.012, the expected cell count for this category was below the minimum value of 5 and it was disregarded.

**Table 5.** Reasons for choosing a university.

| Reasons for Choosing a University | | Foreign Students | | Romanian Students | | Total |
|---|---|---|---|---|---|---|
| | | Count | % | Count | % | |
| University | Count/% | 48 | 30.38% | 110 | 69.62% | 158 |
| | Expected Count | 47.6 | 30.13% | 110.4 | 69.87% | 158 |
| Location | Count/% | 3 | 16.67% | 15 | 83.33% | 18 |
| | Expected Count | 5.4 | 30.00% | 12.6 | 70.00% | 18 |
| Employability | Count/% | 4 | 100.00% | 0 | 0.00% | 4 |
| | Expected Count | 1.2 | 30.00% | 2.8 | 70.00% | 4 |
| Recommendations | Count/% | 5 | 26.32% | 14 | 73.68% | 19 |
| | Expected Count | 5.7 | 30.00% | 13.3 | 70.00% | 19 |
| Total | Count/% | 60 | 30.15% | 139 | 69.85% | 199 |
| | Expected Count | 60 | 30.15% | 139 | 69.85% | 199 |

Another chi-square test of independence was performed to examine the relationship between student origin (foreign or Romanian) and the importance of choosing a university. The relationship between these variables was significant, $X^2$ (3, $N$ = 199) = 7.620, $p$ = 0.022. The effect size, Cramer's V, was small, 0.196 [43]. Romanian students are significantly more likely to quote employability as being important when choosing a university.

As shown in Table 6, the most known association among students (both Romanian and foreign) is AACSB. while the least known is CEEMAN.

**Table 6.** International accreditation agencies and students' awareness of them.

| Institution | Foreign | | | Romanian | | |
|---|---|---|---|---|---|---|
| | Bachelor | Master | Total | Bachelor | Master | Total |
| ARACIS | 0% | 0% | 0 | 34% | 38% | 48 |
| CFA | 28% | 21% | 15 | 17% | 38% | 25 |
| ACCA | 17% | 21% | 11 | 14% | 38% | 21 |
| NIBS | 3% | 8% | 3 | 21% | 0% | 28 |
| CEEMAN | 3% | 0% | 1 | 3% | 0% | 4 |
| CIMA | 8% | 4% | 4 | 3% | 0% | 4 |
| AMBA | 56% | 38% | 29 | 17% | 0% | 22 |
| AACSB | 75% | 96% | 50 | 40% | 0% | 52 |
| EFMD | 8% | 25% | 9 | 2% | 0% | 3 |
| Total enrolled | 36 | 24 | 60 | 131 | 8 | 139 |

Romanian students were asked if they would choose another university if that university had an international accreditation. Of the 139 respondents, 43.9% said yes, while 56.1% said no. This question was only asked as a follow-up question due to the fact that there is no other accredited institution in Romania and no other is pursuing accreditation. We wanted to test the importance of international accreditation as a differentiation factor between two choices. For French students, this question was not asked as it is redundant with more French universities having international accreditation.

Next, we wanted to examine if the internationality of a university and international accreditation have the same underlying concept. As a result, an Exploratory Factor Analysis was performed.

The assumption of the normality of data was tested with a Kolmogorov–Smirnov test for normality. The results show that the data is not normally distributed (Table 7).

**Table 7.** Perception of international accreditation.

| Items | Kolmogorov-Smirnov * | | | Shapiro-Wilk | | |
|---|---|---|---|---|---|---|
| | Statistic | df | Sig. | Statistic | df | Sig. |
| International accreditation means a recognition of the quality of the educational process | 0.242 | 199 | 0 | 0.796 | 199 | 0 |
| International accreditation means I am studying at a top university | 0.234 | 199 | 0 | 0.838 | 199 | 0 |
| The international dimension of a university/business school is very important | 0.363 | 199 | 0 | 0.717 | 199 | 0 |

* Lilliefors Significance Correction.

As a result, a Principal Axis Factoring analysis was performed as suggested by Osborne [44]. A Promax rotation was performed. Although the KMO value was mediocre (KMO = 0.629), Bartlett's Test of Sphericity was significant ($\chi^2(3) = 65.800$, $p <. 001$) and indicates that sample size is adequate for the analysis. The communalities ranging from 0.231 to 0.479 are low suggesting that the three items do not share too much variance. The analysis resulted in a single factor explaining 35.54% of the variance. The reliability of the scale composed of the three items was below the recommended threshold of 0.7 (Cronbach's alpha was 0.613).

Looking at the results of the questionnaire, we noticed that internationalization and international accreditation do play a role in students' choice. This result is not only based on citizenship, but it is something noticeable for all students. Having these results, we wanted to have a look at the benefits of the increase in student population, as well as at the impact of international accreditation in the case of Transylvania Business School.

Next to aspects such as having international accreditations, students are also influenced to choose a faculty based on the paying opportunities.

At the bachelor level, the number of students that benefit from a full-time Bachelor's funded by the state and those that have part-time bachelor tuition remains mainly constant during the academic years. On the other hand, the number of students paying full-time bachelor tuition fluctuates during the analyzed period. In the academic year 2015–2016, exists a peak of 305 fee-paying students, since then the number decreases. Looking at the percentage change we noticed that there is a steady trend of decline representing around 9% but this is due to the demographic changes. The year 2015–2016 represents a strong boost and might be due to the increased advertising of the NIBS accreditation (Figure 2).

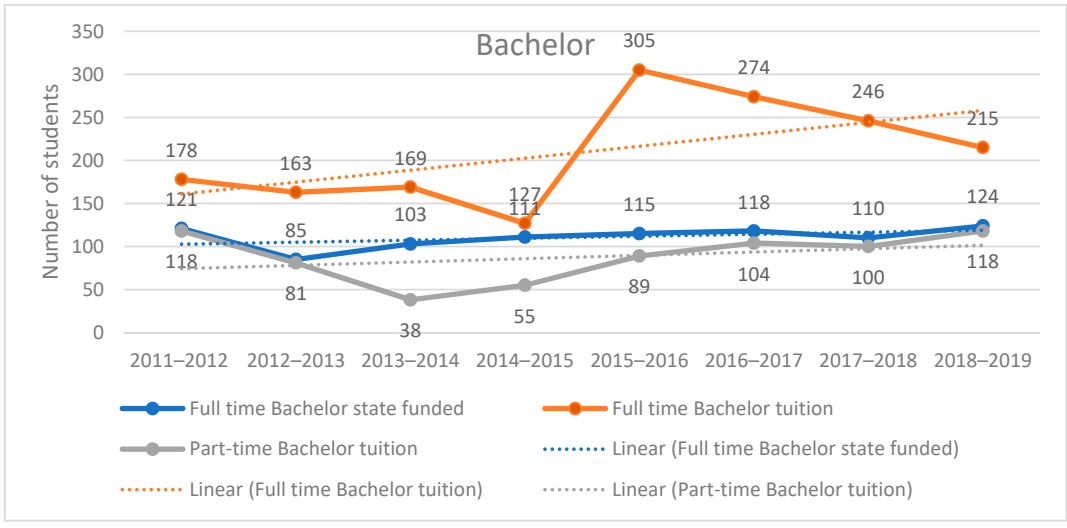

**Figure 2.** Students accepted at Transylvania Business School per academic year at the bachelor level.

At the master's level, the situation looks different. The trend shows a decline in the number of students that benefit from a full-time Bachelor's funded by the state. In the case of students that pay their own tuition, the number remains constant during the analyzed academic years. Similar to the Bachelor level, this can be explained due to the demographic changes happening in Romania (Figure 3).

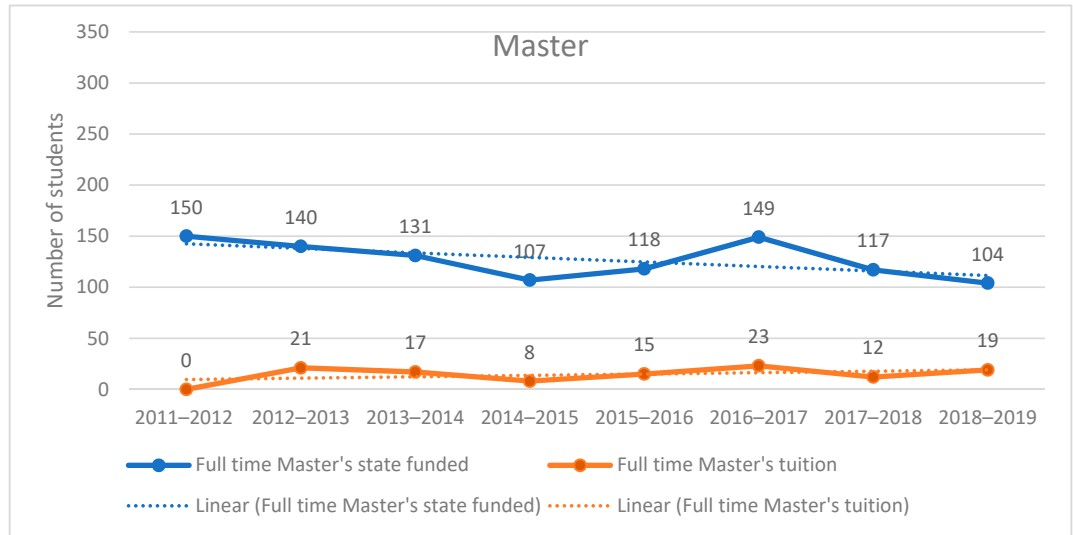

**Figure 3.** Students accepted at Transylvania Business School per academic year at the master level.

Comparing Romanian fee-paying students to the foreign ones, we noticed that the percentage change was on an upward trend and in the last years, the number stabilized. Given the declining number of high school students, foreign students represent a growth market due to the competitive advantages of the Romanian education system. Although Romania is considered a low-cost option for studies among the EU Members States, we consider that on its own, this is not a sustainable competitive advantage and it must be doubled by a high standard of teaching given by accreditation (Figure 4).

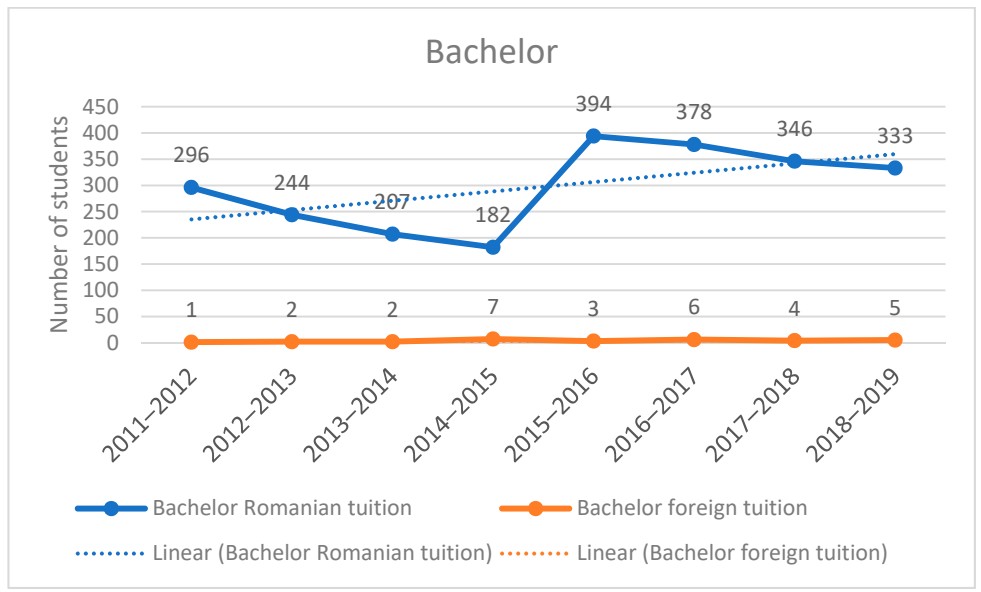

**Figure 4.** Tuition students: Romanian vs. foreign at the bachelor level.

At the master's level, we noticed the same trends, a decline of the domestic students, but a steady increase in foreign students. This result is normal as master's studies are a continuation of Bachelor's studies. We also take into account that a number of bachelor's students do not carry on their studies

because they are tempted by the number of opportunities in the job market (especially during the period when Romania experienced the highest GDP growth in the EU). Another reason for lower numbers at the master level is the fact that this level of studies is considered a specialization in a career and students look for specific programs in fields of economics or business (Figure 5).

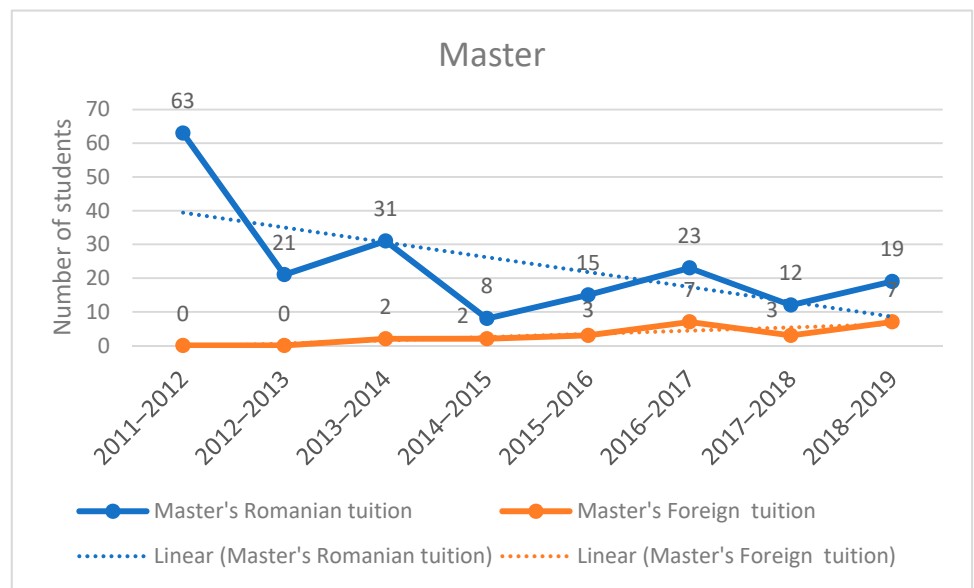

**Figure 5.** Tuition students: Romanian vs. foreign at the master level.

In the case of foreign students coming to study at Transylvania Business School, for both incoming Erasmus students and incoming fee-paying students there is a growing trend. If in the academic year 2011–2012 there were zero students, by 2018–2019 the number increased by 45 Erasmus students and 35 fee-paying students. In percentage change, this represents a constant growth of 26–30%. We noticed that from a financial point of view these fee-paying students are balancing the loss of revenue from the full-time fee-paying students. The number of exchange students could increase if the Erasmus program would receive better funding in the framework of the new EU budget (Figure 6).

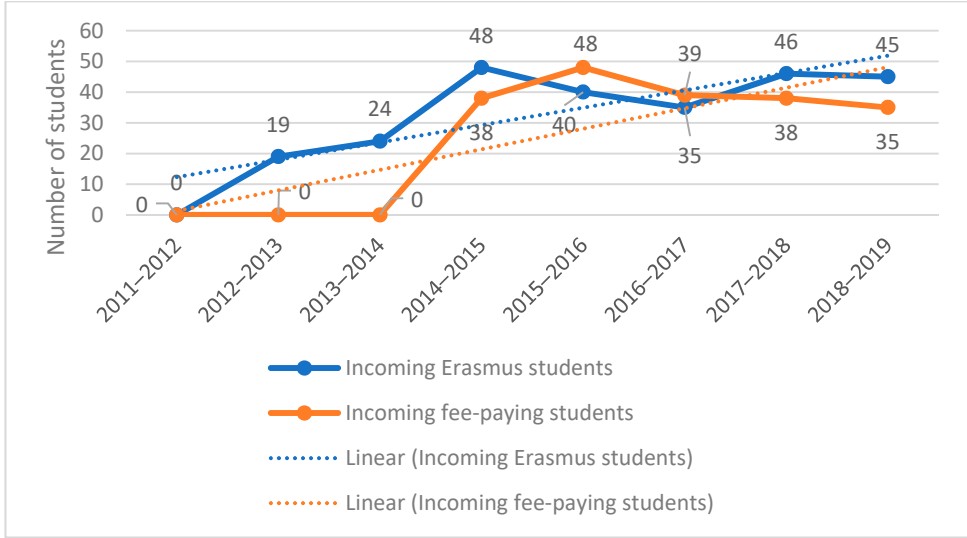

**Figure 6.** Incoming Erasmus students and fee-paying exchange students.

Figures 2–4 and Tables 8–10 show that following NIBS accreditation, the number of students has increased. It is also important to mention that the revenue of the business school has also increased

as a result of this higher number of students. The categories that have had the highest impact on revenue are the foreign tuition students, both bachelor's and master's, and tuition-based exchange students. These students would not have registered if it had not been for the international accreditation. The number is slightly declining because of the increasing number of AACSB accredited institutions and the fact that students are becoming more interested in this type of accreditation. When looking at student numbers we see that the evolution is positive compared to that of the general outlook on the higher education system in Romania. Table 8 shows the evolution of students between 2011 and 2018, based on data obtained from the Ministry of National Education [45], and we can see that universities lost 24.39% of students while Transylvania Business School only lost 8% of fee-paying Romanian students which were compensated by the increase in foreign students.

**Table 8.** Students enrolled in universities in Romania 2011–2018.

|  | 2011–2012 | 2012–2013 | 2013–2014 | 2014–2015 | 2015–2016 | 2016–2017 | 2017–2018 |
|---|---|---|---|---|---|---|---|
| Total students | 539.852 | 464592 | 433.234 | 411.229 | 410.697 | 405.638 | 408.179 |
| % change in total students |  | −13.94% | −6.75% | −5.08% | −0.13% | −1.23% | 0.63% |
| Total foreign students in public universities |  |  |  | 19.231 | 20.577 | 21.556 | 22.768 |
| % change in foreign students |  |  |  | - | 7.00% | 4.76% | 5.62% |

**Table 9.** Costs with AACSB accreditation.

| Fees for Schools Seeking Accreditation | Current Fees | Description |
|---|---|---|
| Membership Fee | 3300 USD | Annual Membership fee |
| Eligibility Application Fee | 2000 USD | One-time fee due following the submission of the eligibility application. |
| IAC or AAC Process Acceptance Fee | 6500 USD | One-time fee due upon the IAC or AAC's acceptance of the eligibility application. |
| Initial Accreditation Fee (Business and Accounting) | 5950 USD | Annual fee due while in the initial accreditation process. The fee is first assessed following the acceptance of the eligibility application. |
| Initial Business or Initial Accounting Accreditation Visit Application Fee | 15,000 USD | One-time fee due following the submission of the initial accreditation application. |
| Deferral Visit Fee | 5500 USD | One-time fee due if the school is placed on a deferral review. |
| Fees for Accredited Institutions | Current Fees | Description |
| Annual Accreditation Fee (Business) | 5950 USD | Annual fees assessed for all business accredited institutions. |
| Annual Accreditation Fee (Accounting) | 3650 USD | Annual fees assessed for all accounting accredited schools in addition to the business fees. |
| Continuing Review Fee (CIR2, FR1, FR2) | 5500 USD | One-time fee assessed if the school is placed on a continuing review. |

Source: AACSB website.

**Table 10.** Cost–benefit analysis.

|  | 2015 | 2016 | 2017 | 2018 |
|---|---|---|---|---|
| Benefits | €53,520 | €52,020 | €59,520 | €68,520 |
| Operating Costs | €24,084 | €23,409 | €26,784 | €30,834 |
| Salary of employee | €16,000 | €16,000 | €16,000 | €16,000 |
| AACSB Accreditation | €5000 | €7000 | €9000 | €20,000 |
| NIBS Accreditation | €500 | €500 | €500 | €500 |
| Total Operating Costs | €45,584 | €46,909 | €52,284 | €67,334 |
| Cash flow from operations | €7936 | €5111 | €7236 | €1186 |

Source: Authors' own calculations.

The NIBS accreditation process was estimated to cost 6000 Euro, which included the accreditation fee and the membership fee starting from 2015 and the participation fee at different events. As mentioned before, the AACSB accreditation is not a benchmark but a process accreditation and thus, the cost cannot be estimated until the end. However, by looking at the expenses that occurred in connection with AACSB accreditation, according to their webpage, a business school could expect the following costs (Table 9).

Besides these costs, a school could expect to spend money on participation in seminars and conferences and on the implementation of the accreditation standards. Depending on the development of the business school and how ready the school is for this purpose, this cost could prove to be high.

To see if the decision of Transylvania Business School was a good one and if it is sustainable in the long run, we wanted to apply a cost–benefit analysis. This analysis was carried out for the years 2015–2018 as the international accreditation process started in 2015 with NIBS and 2018 was chosen because it was the last financial year where the authors had the data. Additionally, 2018 also represents the year when the school obtained the Eligibility Application fee, which meant a substantial cost, and starting with that process which includes a 5950 USD cost above the membership. Thus, we can expect that costs are steady from that point in time excluding the moment of the Initial Accreditation Visit.

For the cost–benefit analysis, we followed Berk and DeMarzo [45] and used the Net Present Value as an investment decision tool. The standard formula for the Net Present Value is $NPV = -I + \sum_{i=1}^{n} \frac{CFW^i}{(1+c)^i}$. where $I$ is considered the initial investment, $CFW$ the operating cashflow of a certain year and $c$ the cost of capital. When looking at the formula we took into account that the initial investment is irrelevant as the cost for the accreditation is not a one-time fee but a cost that is split throughout the accreditation journey as can be seen from the table which presents the operational cash flow.

In order to look at the operating cash flow generated by the international accreditations, we looked at the new revenue streams and the expenses. In these revenue streams, we included 50% of the international students both at the master and bachelor levels, as well as the international fee-paying students. All numbers that make the basis for these calculations have been provided by the management of the school under Law 544 on the freedom of information law of 12 October 2001. For this calculation we have included all registered tuition fee foreign students, and not just the ones being admitted in that year in the school and a 3000 Euro/year tuition for each student was taken into account. As the international fee-paying students stay only one semester or one year maximum, we took the yearly intake and a 700 Euro/student fee was calculated.

For the benefits, we estimated only 70% of the actual revenue from tuition, due to the fact that some foreign students do not actually pay their tuition, or because other miscellaneous fees have occurred. In the operating expenses, we have included the costs directly associated with the accreditation fee, the average salary of a person working in the administration in a university in Romania (16,000 Euro/year) and a supplementary lump sum of 45% of the revenue which was calculated based on experience. Of this lump sum, 23 percentage points represent the administration fee of the university and the other 22 percentage points represent costs at the business school level (Table 10).

As there are no initial costs, we have taken $I = 0$. As the university is a public one and investments are limited to state investments, we took for the cost of capital as the 10-year yield of treasury bonds issued by the Romanian state, $c = 3.99\%$.

By applying all our information, we have the following equation:

$$\text{NPV} = \frac{€7936}{(1+3.99\%)^1} + \frac{€5111}{(1+3.99\%)^2} + \frac{€7236}{(1+3.99\%)^3} + \frac{€1186}{(1+3.99\%)^4}$$
$$NPV = €19,806.64 \tag{1}$$

A project is successful when the Net Present Value is positive. It means that the cash generated by the project is higher than the initial investment. The discounting of the cash flows from years 1 to n makes sure that the change in the currency value, most often inflation, is also taken into account. Thus, the entire decision is to be taken in year 0 which is the moment when the management has to decide upon a project.

The results of our cost–benefit analysis show that the entire process can be considered a success and is in line with our third hypothesis. The NIBS international accreditation and the AACSB accreditation processes have resulted in a marginally high and positive NPV. Obviously, we could introduce into discussion the fact that as there was no initial investment, a simple cost–benefit analysis could have shown similar results. By a simple cost–benefit analysis, we refer to calculating the cost for the

accreditation process and the benefits which arose. The results of such a basic analysis would be the operating cash flow which was calculated in Table 10. We consider that the Net Present Value a better option because it includes the discounting of future cash flows.

Our analysis is just part of the cost–benefit analysis of an international accreditation process, and we are sure that once the school will obtain the accreditation, the use of the accredited logo will lead to an increase in student numbers.

## 5. Discussion

The current study presents the results of a cost–benefit analysis of the international accreditation process. The decision to engage in an accreditation process is not easy and there are a lot of factors that impact this choice [46,47]. The decision is not only based on the quality and meeting the standards, but it can also be a financial decision. With lower numbers and lowering of public finances, opting for such a costly process is difficult and many universities are taking their time to decide. Our study deploys a cost–benefit analysis on a small- to medium-sized business school located in Romania which is relying partly on public funding. The analysis shows that international accreditations are a costly process, but their benefits are higher than the costs. We included in the research only the revenue that can be directly linked with international accreditation—international students and we did not take into account that the AACSB and NIBS would attract Romanian students. However, our study shows that Romanian students are aware of the importance of internationalization. It does not show significant differences between Romanian and French students or students from other universities that were in exchange programs at the two business schools. The only significant difference is in the awareness of institutional accreditation agencies. While more than 75% of the students from EM Normandie have heard about AACSB, less than 10% know about NIBS. Among the Romanian students, only 21% know about NIBS and 40% have heard about AACSB. This percentage is explained because of the promotion of NIBS accreditation and the AACSB process in which Transylvania Business School is involved. Additionally, 34% and 38% of the Romanian students have heard about ARACIS which states the importance of the national accreditation, but the fact that AACSB is at 40% makes the authors believe that the impact of having an AACSB quality seal will be important. We know that our study is limited due to the focus on one school and the fact that the business school did not need to invest vast amounts of money in setting up the standards for the AACSB accreditation. The findings of our study could be improved if the response rate of students was increased or if all the students were required to fill in the questionnaire at the beginning of the academic year. The cost–benefit analysis is also limited because the AACSB accreditation is still an ongoing process. Thus, from a marketing point of view, its impact on the revenues is still limited because the AACSB members cannot promote the accreditation process itself and must only advertise the membership in the association. Also, the fact that Transylvania Business School has not received the accreditation yet could mean that more investments should be made in ensuring that all the required standards are met.

## 6. Conclusions

In the context of the discrete, yet harsh competition among the universities in our country for attracting students—a task that has become increasingly difficult in the last years because of the demographic trends, emigration process, etc.,—the purpose of our study was to see if international accreditations have an impact on students' choice of university, if these are sustainable, and worth pursuing. Moreover, nowadays this competition is also taking place at the international level; therefore, it would be important not just to focus on attracting local students, but also international students, and in order to do so, we believe that international accreditations are vital. Consequently, we tested the impact and the sustainability of international accreditations using a cost–benefit analysis and we looked deeper into it by analyzing the determinants of students' choice in universities. Two out of three initial hypotheses were validated by our study. Our results prove that the international dimension does not play such an important role for domestic students, but it is important for foreign students. Linked

to this, looking at the Net Present Value we have seen that over a period of four years the international accreditation process has been a success, as the NPV is higher than zero and marginally high. Not only is the Net Present Value positive but we see that in each of the four years the project has generated positive cash flows. As mentioned before in our analysis, we only took international students into account but we noticed that international accreditation is also more and more important to domestic students as shown in previous studies like [23,32,34]. As far as the students' choice is concerned, the results are different from the ones of [25] but are in line with the studies of Tamtekin Aydın [20] and Winter and Chapleo [23], which show that the content of the program plays an important role. Tamtekin Aydın [20] presents the content of the program as among the least important reasons, but our study shows that it is the most important one, while family is among the last ones. Additionally, we noticed that international accreditation is playing an important role, as it is also mentioned in the reasons for choosing the program, as well as in determining the quality of the program. Students' employability is not seen as a major reason when selecting a university, but it is seen as an important element for choosing a program.

As a future research plan, our intention is to evaluate the results after the business school finishes the accreditation process and receives the AACSB membership and, further on, to carry out a comparative study with a similar business school that has chosen not to opt for international accreditation.

**Author Contributions:** Conceptualization, D.P. and F.S.D.; methodology, D.P and C.F.; software, C.F.; validation, I.A.N., D.P., and F.S.D.; formal analysis, C.F.; investigation, D.P.; resources, I.A.N.; data curation, C.F.; writing—original draft preparation, D.P.; writing—review and editing, D.P.; visualization, C.F.; supervision, I.A.N. and F.S.D.; project administration, D.P. All authors have read and agreed to the published version of the manuscript.

**Funding:** This research received no external funding.

**Conflicts of Interest:** The authors declare no conflict of interest.

## Appendix A

Questionnaire delivered to students.

**Question 1**: Please select your gender.

a. Male

b. Female

**Question 2**: Please enter your email (non-mandatory question).

**Question 3**: Please enter the university where you are currently enrolled.

**Question 4**: You are a student in:

a. 1st year bachelor's

b. 2nd year bachelor's

c. 3rd year bachelor's

d. 4th year bachelor's

e. 1st year master's

f. 2nd year master's

**Question 5**: What was the reason for choosing the university/business school where you are currently enrolled

a. city

b. tradition of the business school/university

c. the program where I am enrolled

d. friend's recommendation

e. international accreditation of the business school/university

f. recommendation of the family

g. employability chances

**Question 6**: What is in your view the most important element when choosing a university/business school city?

a. content of the program

b. city

c. employability after graduation

d. international relations

e. national accreditation

f. international accreditation

**Question 7**: Please tick the institutions that you have heard of.

a. AMBA

b. AACSB

c. EFMD

d. CEEMAN

e. NIBS

f. ACCA

g. CIMA

**Question 8**: Please evaluate the following statement: an international accreditation means a recognition of the quality of the educational process.

| Fully disagree | 1 | 2 | 3 | 4 | 5 | Fully agree |
|---|---|---|---|---|---|---|

**Question 9**: Please evaluate the following statement: an international accreditation means that I am studying at a top university.

| Fully disagree | 1 | 2 | 3 | 4 | 5 | Fully agree |
|---|---|---|---|---|---|---|

**Question 10**: Please evaluate the following statement: the international dimension of a university/business school is very important.

| Fully disagree | 1 | 2 | 3 | 4 | 5 | Fully agree |
|---|---|---|---|---|---|---|

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
