# Peer review of "The Sustainability of International Accreditations and Their Impact on Students’ Choices in Selecting the Universities"

_sustainability, doi:10.3390/su12166480_

Round 1

Reviewer 1 Report

The paper explores the sustainability of international accreditation through the dual lens of cost and the impact of accreditation upon students’ choice of universities in which to enrol, within the context of French and Romanian students in particular.

It provides a useful overview of the state of accreditation systems

The study used a 7 question questionnaire for students

There should be a clearly articulated list of research questions that includes the 3 Hypotheses

Tables 3, 4, 5 would be more informative and comparative using percentages rather than numbers of students.

Tables 8, 9 ,10 would be better represented as charts for a quicker visual comparison of the results.

Section 4 Discussion is actually the Conclusion - and should be divided into two sections - limitations of the study, and conclusions.

The questionnaire should be attached as an appendix, and there should be a discussion on the design of the questionnaire and the ethics consent process for students, as well as detail of how the questionnaires were administered.

Author Response

First of all we want to thank the reviewer for the valuable input. We believe that after extensive reviewing and rewriting of the article it's research impact has grown and will contribute to the literature. 

1. There should be a clearly articulated list of research questions that includes the 3 Hypotheses

Response:

A clearer research question was introduced in line 96. "The main question that our study wants to answer is linked to the financial sustainability of an international accreditation and the profitability of such endeavour. Linked to this the current study wants to see if the international accreditation is a factor for student’s choice in programmes and if there are regional difference between France and Romania"

2. Tables 3, 4, 5 would be more informative and comparative using percentages rather than numbers of students.

Response: Indeed looking at the tables we have noticed that for the purpose of the future references of this study it might be better to include percentages. We have included both numbers and percentages in the aforementioned tables. 

3. Tables 8, 9 ,10 would be better represented as charts for a quicker visual comparison of the results.

Response: Indeed it is difficult to follow all the numbers. The tables were redesigned as a charts and we have divided them into bachelor and master's for a clearer view as the numbers were very different. 

Section 4 Discussion is actually the Conclusion - and should be divided into two sections - limitations of the study, and conclusions.

Response: Thank your very much for the suggestion. This suggestion was also made by another reviewer and thus we have created two distinct sections -. "Discussions" where we have included a discussions regarding the issue and our findings and limitations. In the "Conclusion" sections we have introduced our key findings and future research plans. 

The questionnaire should be attached as an appendix, and there should be a discussion on the design of the questionnaire and the ethics consent process for students, as well as detail of how the questionnaires were administered.

Response: Thank you for this suggestion.In the Data Source and Method we have included a discussion related to the design and delivery of the questionnaire in the text starting with line 418. We have also attached the questionnaire as an Appendix.  

Reviewer 2 Report

The authors investigated whether or not the international accreditations for business schools could be the main reason of students’ decision in determining their university selection. The authors conducted a survey and verified that the international accreditations for business schools did play an essential role of making a decision for their university selection. Overall, the topic of this paper is interesting, and the research methodology and results are acceptable. However, this paper consists of some shortcomings. Before this paper can be further considered for publication, this paper needs a major revision. Below, I describe my main concerns and suggestions.

  • In the abstract section, the authors said “… we have tested it by applying a questionnaire to 400 business and economics students…”, but in the Results section the authors said “The sample was comprised of 60 foreign students and 139 Romanian students.” That is, there were 199 students in total rather than 400 students. So, where were the other 201 students? They were not consistent in the paper. Did the authors make a mistake? Or, did the authors mean that they sent out the survey to 400 students but only 199 students responded to the survey? The authors should clarify it clearly; otherwise, it would confuse readers.  
  • The authors need to create an Appendix section to show all questions in the questionnaire. So, readers could get the whole picture and realize what the authors wanted to investigate.
  • The size of the current Introduction section is too long. Indeed, the current Introduction section includes “introduction” and “literature review”. The authors need to split the current Introduction section into two sections: Introduction and Literature Review. In the introduction section, the authors describe the background of this issue, motivation of writing this paper, why this issue is so important to be raised for discussion, and develop research questions. In the literature review section, the authors use past studies of this issue or/and related issue to formulate the theoretical background for this issue, and then develop research hypotheses.
  • The current second section “Materials and Methods” should be re-named to “Data Source and Research Methods”. The authors wrote three hypotheses in the current second section, which were placed incorrectly. As mentioned earlier, the research hypotheses should be placed in the literature review section. That is, after formulating theoretical background, the authors develop research hypotheses based upon the theoretical background. So, your research hypotheses have a strong theoretical background to sustain. In addition, the words of current three hypotheses are too long and not clear. Each hypothesis should be short, concise, and clear in one sentence; thus, readers can easily read and immediately understand what the authors want to test.
  • For Tables 8, 9, and 10, I suggest that the authors also draw diagrams so that readers can easily see the trend. In addition, the authors may need to show the increasing rate yearly in these tables, so readers can easily see the change yearly.
  • The authors need to explain the source of those revenue data. For example, you need explain to readers which unit of the university provided these revenue data to you for this research.
  • This paper does not have a conclusion section. Any paper must need a conclusion section. As I read the current “Discussion” section of the paper, it reads like conclusion. Therefore, the authors need to rewrite the discussion section and create a conclusion section. In the discussion section, the authors need to provide more thoughtful discussions regarding this issue and their findings and what their results can imply. The authors also can discuss their research limitation in the discussion section. In the conclusion section, the authors can conclude their key findings and future research plan regarding this research.

Author Response

First of all we want to thank the reviewer for the valuable input. We believe that after extensive reviewing and rewriting of the article it's research impact has grown and will contribute to the literature. 

  • In the abstract section, the authors said “… we have tested it by applying a questionnaire to 400 business and economics students…”, but in the Results section the authors said “The sample was comprised of 60 foreign students and 139 Romanian students.” That is, there were 199 students in total rather than 400 students. So, where were the other 201 students? They were not consistent in the paper. Did the authors make a mistake? Or, did the authors mean that they sent out the survey to 400 students but only 199 students responded to the survey? The authors should clarify it clearly; otherwise, it would confuse readers.  

Response: Indeed this was an error on our part. The questionnaire was sent to 729 students from both business schools and participation was voluntary. We have received 400 answers but only 199 were questionnaires that were consistent and where all questions were validated by the authors. The issues arised was corrected and we have introduced a clarifying note in the data source and method section starting with line 418.  

  • The authors need to create an Appendix section to show all questions in the questionnaire. So, readers could get the whole picture and realize what the authors wanted to investigate.

Response: Thank you very much, the introduction of an Appendix was suggested also by another reviewer and we have included the Questionnaire in the end of the paper as an Appendix. 

  • The size of the current Introduction section is too long. Indeed, the current Introduction section includes “introduction” and “literature review”. The authors need to split the current Introduction section into two sections: Introduction and Literature Review. In the introduction section, the authors describe the background of this issue, motivation of writing this paper, why this issue is so important to be raised for discussion, and develop research questions. In the literature review section, the authors use past studies of this issue or/and related issue to formulate the theoretical background for this issue, and then develop research hypotheses.

Response: Thank you for the suggestion. Indeed the introduction was too long. We have created to separate sections - "introduction" and "literature review" and we have followed your suggestions and the ones from another reviewer and have included more recent studies in the last section. 

  • The current second section “Materials and Methods” should be re-named to “Data Source and Research Methods”. The authors wrote three hypotheses in the current second section, which were placed incorrectly. As mentioned earlier, the research hypotheses should be placed in the literature review section. That is, after formulating theoretical background, the authors develop research hypotheses based upon the theoretical background. So, your research hypotheses have a strong theoretical background to sustain. In addition, the words of current three hypotheses are too long and not clear. Each hypothesis should be short, concise, and clear in one sentence; thus, readers can easily read and immediately understand what the authors want to test.

Response: Thank you for the suggestion. The Materials and Method was renamed "Data Source and Research Methods" and we have moved the hypothesis, which are now clearer formulated to the introduction section starting with line 96. 

  • For Tables 8, 9, and 10, I suggest that the authors also draw diagrams so that readers can easily see the trend. In addition, the authors may need to show the increasing rate yearly in these tables, so readers can easily see the change yearly.

Response: The change from tables to figures was suggested by another reviewer and we thank you for this suggestion. We have changed the format, and we have included charts but we have kept the tables to show the yearly change, as the charts were too crowed for several informations. 

  • The authors need to explain the source of those revenue data. For example, you need explain to readers which unit of the university provided these revenue data to you for this research.

Response: Thank you so much for pointing out the revenue section of our data. Looking at the numbers we noticed an error in the calculation of the Benefits and we corrected table 13 and the NPV formula. The data was obtained from the management of the business school as all of the information is also public data available for every individual. We have explained in a footnote the values (tuition fee for the full time international students and fee-paying exchange students). 

  • This paper does not have a conclusion section. Any paper must need a conclusion section. As I read the current “Discussion” section of the paper, it reads like conclusion. Therefore, the authors need to rewrite the discussion section and create a conclusion section. In the discussion section, the authors need to provide more thoughtful discussions regarding this issue and their findings and what their results can imply. The authors also can discuss their research limitation in the discussion section. In the conclusion section, the authors can conclude their key findings and future research plan regarding this research.

Response: Thank your very much for the suggestion. This suggestion was also made by another reviewer and thus we have created two distinct sections -. "Discussions" where we have included a discussions regarding the issue and our findings and limitations. In the "Conclusion" sections we have introduced our key findings and future research plans.

Reviewer 3 Report

The article presented is of interest to the scientific community, but the authors must attend to the following observations for the improvement of the manuscript:

-Authors should pay attention to the journal's format rules.

-Authors must structure the Materials and Method section in subsections. In addition, they must specify the investigative procedure carried out in order to replicate the study. They must also detail the research methodology used.

-The presentation of the results must be improved. The tables must be cited in the text and placed after the paragraph of explains them.

-The discussion section is very short. This section together with the conclusions are the most relevant sections of a scientific article because they contribute new knowledge to the scientific community. Therefore, it must be modified in depth.

-The work has very superficial limitations. Authors should delve into them. In addition, they must present the future lines of study, derived from the present work.

-Reviewing the literature on the state of affairs is not enough. And it is not current either. The final list of references must conform to the standards of the journal.

Author Response

First of all we want to thank the reviewer for the valuable input. We believe that after extensive reviewing and rewriting of the article it's research impact has grown and will contribute to the literature. 

-Authors should pay attention to the journal's format rules.

Response: We hope that with the suggestions given by the reviewers and the editorial team we have respected the rules and guidelines of the journal. 

-Authors must structure the Materials and Method section in subsections. In addition, they must specify the investigative procedure carried out in order to replicate the study. They must also detail the research methodology used.

Response: As suggested by two other reviewers we have changed this section in Data Source and Method. We have indicated the procedure of the study starting with line 418.  

-The presentation of the results must be improved. The tables must be cited in the text and placed after the paragraph of explains them.

-The discussion section is very short. This section together with the conclusions are the most relevant sections of a scientific article because they contribute new knowledge to the scientific community. Therefore, it must be modified in depth.

Response: The discussion section was split into "Discussion" and "Conclusions" as suggested by another reviewer. Both sections were expanded and we hope that know it includes the desired information. 

-The work has very superficial limitations. Authors should delve into them. In addition, they must present the future lines of study, derived from the present work.

Response: The limitations of the study were reviewed and better presented and are visible starting with line 550. 

-Reviewing the literature on the state of affairs is not enough. And it is not current either. The final list of references must conform to the standards of the journal.

Response: Thank you very much for this suggestion. Looking better at the style guidelines we have notice that an EndNote style for MDPI is available and we have formated the paper using EndNote X9 with the MDPI Style. Also, we have looked at the references and we have expanded our literature review and included several other journals and hope that now we meet the standards of this journal. 

Round 2

Reviewer 2 Report

The authors have followed my comments and suggestions to revise the paper. I am satisfied with their revision.

Author Response

We thank you very much for your suggestions. Your feedback has increased the value of our paper! 

Reviewer 3 Report

Although the authors have made an effort to improve the manuscript. There are still other aspects to improve. The authors have not adapted the tables to the standards required by the journal and do not adequately present the results. Each table and figure must have an explanatory text with a description that does not reiterate the information. Authors cannot post multiple tables and figures in a row. They need to improve the presentation of the results.

Author Response

Thank you very much for pointing out some of the key elements that we did not take into account. We have changed the tables and figures and we have adapted them according to the template provided by the journal. 

We also apologise for including several tables and figures one after the other but the change was suggested by one of the reviewers. Nevertheless, we have now improved the presentation of the results and we have included the vital information which was in some the tables in the text. We hope that the quality of the presentation of the results has much improved. We have highlted the connection between our initial hypothesis and our results and we also included some other discussions in our analysis. 

We have included a revised version of our paper that tracks the changes done since the last revision. 

Round 3

Reviewer 3 Report

The authors have made the observations offered to improve the manuscript. Therefore, I recommend acceptance of the study.